# Intervention Activities Associated with the Implementation of a Comprehensive School Tobacco Policy at Danish Vocational Schools: A Repeated Cross-Sectional Study

**DOI:** 10.3390/ijerph191912489

**Published:** 2022-09-30

**Authors:** Anneke Vang Hjort, Mirte A. G. Kuipers, Maria Stage, Charlotta Pisinger, Charlotte Demant Klinker

**Affiliations:** 1Health Promotion Research, Steno Diabetes Center Copenhagen—Copenhagen University Hospital, 2730 Herlev, Denmark; 2National Institute of Public Health, University of Southern Denmark, 5230 Odense, Denmark; 3Department of Public and Occupational Health, Amsterdam UMC Location University of Amsterdam, 1105 AZ Amsterdam, The Netherlands; 4Health Behaviors & Chronic Diseases, Amsterdam Public Health Research Institute, 1081 BT Amsterdam, The Netherlands; 5The Danish Cancer Society, 2100 Copenhagen, Denmark; 6Center for Clinical Research and Prevention, Capital Region, 2000 Frederiksberg, Denmark; 7The Danish Heart Foundation, 1120 Copenhagen, Denmark

**Keywords:** school tobacco policy, implementation, implementation fidelity, implementation outcomes, vocational schools, smoke-free school hours

## Abstract

School tobacco policies are often poorly implemented, which may explain their limited effectiveness. Further, constructs to measure implementation outcomes of school tobacco policies are missing. The Smoke-Free Vocational Schools intervention was designed to stimulate the implementation of a comprehensive school tobacco policy into routine practice. This study (1) developed implementation fidelity outcomes measures for the school tobacco policy and (2) examined associations between intervention activities and implementation fidelity at two time points. We applied a repeated cross-sectional survey study design across seven schools: the first time point was >5 months after the policy was established and the second time point > 14 months after policy establishment. The dependent/outcome variables were four binary fidelity domains as well as a total score across domains. A total of six intervention activities were measured among either students (e.g., new school-break facilities) or staff/managers (e.g., a joint workshop before policy implementation). Associations were analyzed separately for students and staff/managers using generalized linear mixed models, adjusted for confounders. A total of *n* = 2674 students and *n* = 871 staff/managers participated. The total implementation fidelity scores increased over time among both students and staff/managers. Three intervention activities were consistently associated with the total implementation fidelity score, including: new school-break facilities (B_T1_ = 0.08, 95% CI = 0.03; 0.12; B_T2_ = 0.07, 95% CI = 0.04–0.50), the joint workshop before policy implementation (B_T1_ = 0.13, 95% CI = 0.02; 0.25; B_T2_ = 0.13, 95% CI = 0.004; 0.24), and internalization of fixed procedures for enforcement (B_T1_ = 0.19, 95% CI = 0.13–0.26; B_T2_ = 0.16, 95% CI = 0.13–0.26). These findings can be applied by schools and other actors in practice. The developed implementation fidelity outcomes measures can be applied in future research on school tobacco policies.

## 1. Introduction

School tobacco policies (STPs) specify for whom, where, and when smoking is prohibited during the school day as well as the consequences for violating the rules. Implementing STPs is a key strategy to reduce smoking rates among adolescents across Europe [1]; targeting students at vocational schools is especially important as the smoking prevalence is high among individuals with lower levels of education [2]. In Denmark, the daily smoking prevalence for vocational students was 29% in 2020 (compared to 9% among peers in academic-oriented upper-secondary education) [3,4]. Among the general Danish population, 13% smoked cigarettes daily in 2020 [5].

A landmark review by Galanti et al. (2014) synthesized decades of research on STPs and found inconclusive evidence of the effectiveness, largely due to inadequate implementation as well as insufficient and inconsistent measurement across studies [6]. Inadequate STP implementation is alarming as it may have a reverse impact on students’ smoking behavior and beliefs [7]. For example, lack of enforcement may lead to young people internalizing beliefs that the schools’ anti-smoking messages are overstated and smoking is not so bad [7]. Insufficient measurements may result in misinterpretation of effects, as limited effects might not be due to a weakness in the STP design, but due to poor implementation [8,9]. In addition, inconsistent STP measurement terminology makes it difficult to synthesize evidence.

We identified two interrelated research gaps associated with the implementation of STPs: (1) evidence on how schools can facilitate successful implementation of STPs and (2) a valid set of STP implementation outcome measures are needed. Below, we elaborate on these research gaps and how this study aims to address them.

In this study, implementation is conceptualized as the process of integrating an intervention into routine practice in a specific setting [10]–STPs in vocational schools. Successful implementation is achieved when the intervention ‘disappears’ from view, that is, is normalized [11,12,13]. As for STPs, school staff must enforce the policy as a normal part of their professional practice and students experience the policy as a normal part of everyday life at school. Implementation outcomes describe the result of an implementation effort [14]. Despite being mandated by law in many European countries [1,15], STP implementation outcomes are often insufficient.

Most studies on STPs focus on effectiveness (i.e., the smoking behavior of students) without careful consideration of how implementation outcomes are measured. That is, previous STP measures have tended to cover some aspects relevant to implementation, for example, enforcement, but not all relevant domains [6]. Although several studies [16,17,18,19,20,21,22] have reported the lack of an STP measure that reflects the actual implementation as a methodological shortcoming, the problem remains unsolved.

The concept of implementation fidelity is the most used implementation outcomes measure [14]. The concept usually comprises four domains: adherence to the program, dose (the amount of program delivered), quality of program delivery, and participant responsiveness [23]. Drawing on the fidelity literature, one previous study operationalized an implementation fidelity measure to capture the degree to which a STP was implemented as intended [24].

As STP implementation must happen among both staff/managers and students, recent studies [25,26] have specified the need to triangulate the data sources. Most studies [16,17,18,21,24,27,28,29,30,31,32,33] that included data from staff/managers used only one respondent (usually the school principal or the school health officer) to assess whether the school had an STP and if so, which one. However, the perspective of one person does not reflect actual implementation [34]. To the best of our knowledge, no studies have applied a representative sample of staff/managers to assess STP implementation.

In addition, several studies [21,25,35,36,37] have highlighted the need to examine STP implementation over time. Yet, only one study [35] has examined the impact of a STP–before and after the establishment of the policy.

In this study, we address the described issues by (1) developing implementation outcome measures based on the fidelity concept and empirical evidence and by (2) measuring the implementation outcomes among representative samples of staff/managers and students over time.

Previous research has identified several facilitators for the successful implementation of STPs. The SILNE-R project [38] investigated the implementation of STPs across seven European countries that were synthesized into recommendations. These recommendations include schools establishing clear rules as well as communicating and discussing the purpose and legitimacy of the policy [39]. In addition, schools must explicate the roles and responsibilities of staff, ensure consistent enforcement with a progressive application of disciplinary measures and establish support for students who struggle or refuse to stop smoking during school hours [39]. A recent Danish qualitative study in vocational schools found similar results and highlighted the need to prepare staff and managers before establishing the policy, that is, to develop organizational readiness for change [40]. The Danish study also demonstrated that establishing new school-break facilities to replace social smoking might ease implementation [40]. Yet, no intervention studies have investigated whether or which actions lead to better STP implementation outcomes.

To address this research gap, we developed the Smoke-Free Vocational Schools intervention [41]. The intervention integrated the evidence on facilitators discussed above into activities tailored to stimulate the implementation of a comprehensive *smoke-free school hours* policy. The intervention activities are described in the Methods section. The policy stipulates that smoking and use of other tobacco products are prohibited during school hours for students, staff, and visitors–inside and outside school premises [41].

In Denmark, as of 31 July 2021, a smoke-free school hours policy has been mandated by law and applies to all educational institutions with students under the age of 18 (including vocational schools). Yet, the challenge of implementing the policy into routine practice remains.

### Aim

This study aimed to assess associations between intervention activities and implementation fidelity of the smoke-free school hours policy from the perspectives of both students and staff/managers at two time points. As a prerequisite and subordinate aim, we have developed a set of STP implementation outcome measures for both respondent groups.

## 2. Materials and Methods

The STROBE checklist for cross-sectional studies was used in the reporting of the study (see Appendix A).

### 2.1. Setting

In Denmark, vocational education and training is an upper-secondary education for a specific industry or trade, for example, carpenter, chef, or hairdresser, and is characterized by a combination of in-school education and workplace training. The age of students varies; some students enroll after completing lower-secondary school at age 15–16, while others enroll later in adult life. Approximately 60% are under the age of 24 (mean age 23.9) [42]. The educational program is divided into four main subject areas: (1) care, health, and pedagogy; (2) Administration, commerce, and business service; (3) Technology, construction, and transportation; (4) Food, agriculture, and hospitality. These subject areas cover 100+ different vocational education programs. Students attend either the normal or the higher educational program; the latter has more theoretical in-school education and, after completion, it provides access to tertiary level education (in addition to a vocational education). Of the total Danish population, 29% held a vocational education diploma as their highest degree in 2020.

### 2.2. The Smoke-Free Vocational Schools Intervention

The Smoke-Free Vocational Schools intervention project aimed to support schools in implementing the smoke-free school hours policy. The intervention took place during 2018–2020 across seven schools that represent all main subject areas as well as three (out of five) geographical regions. The seven intervention schools are considered representative of all Danish vocational schools [41].

The intervention period was approximately one year per school. During the first six months, activities were delivered to stimulate organizational readiness. Then, the policy was established, and during the following six months, activities were delivered to stimulate implementation into routine practice. Table 1 shows the purpose of each of the intervention activities and whether the activity primarily targeted students or staff/managers. For a more comprehensive description of the intervention and program theory, see Hjort et al. (2021) [41].

The project has been reported to the Capital Region of Denmark’s legal center for personal data, reference number VD-2018-485. Given that this study is not a clinical trial, no further ethical approval was needed according to Danish legislation [43].

### 2.3. Study Design

To assess associations between intervention activities and implementation fidelity of the STP, we applied a repeated cross-sectional survey design with two time points (T1 and T2) across the seven intervention schools. This design was appropriate and pragmatic as we (1) aimed to investigate implementation over time and (2) were not able to follow up on the same individuals over time. The first time point was at least five months after the policy had been established, while the second time point was at least 14 months after policy establishment. The exact timing of data collection is shown in Figure 1. Vocational schools in Denmark were closed due to COVID-19 from March 2020 to May 2020 and from the end of December 2020 to March 2021. Data collection was not executed during lockdowns.

Implementation fidelity was assessed among students and staff/managers at both time points. The intervention activities were assessed at either student level or staff/manager level, depending on which respondent group the activity primarily targeted (see Table 1). All intervention activities were assessed at both time points with the exemption of the joint workshop before policy implementation. At most of the schools, the joint meeting took place 1–3 months before policy establishment and was surveyed approximately 2 weeks before the policy was established.

### 2.4. Data Collection

At both time points, we collected information on intervention activities and implementation fidelity using electronic surveys. At staff/manager level, the surveys were distributed to all school staff and managers using the schools’ e-mailing lists. The response rates were 47% at T1 and 50% at T2.

At student level, electronic surveys were completed in the classroom during school hours and under the supervision of a researcher and a teacher: first, the researcher explained the study purpose and answered questions, and then the students completed the questionnaire. Due to COVID-19, some schools did not allow the researchers to access the classroom. In those cases, an online video platform (e.g., Microsoft Teams) was used to introduce the study and supervise data collection (while the students attended school as normal). All vocational students who attended in-school education (i.e., not workplace training) at the time of data collection were eligible to participate. An estimated 95% of the students in the participating school classes completed the questionnaire, regardless of whether they received in-person or online assistance. It is not possible to calculate an exact response rate among the total student population who attended in-school education, as this varies from day to day, and the schools do not have this estimate.

Staff/managers received written information about the study scope, voluntariness, and processing of personal data in the email invitation with the link to the electronic survey. Students received the same written information, and the researchers also explained it to them orally. All participants gave active informed consent before entering the survey.

More information about the questionnaires, pilot testing, and validity, as well as on data collection is available elsewhere [41].

### 2.5. Study Population

Respondents were excluded if they had missing values on any implementation fidelity variables. An exemption, however, is random missing responses on one fidelity item at school 1 at T1 that were due to a survey-technical malfunction (*n* = 102). The total included study population at T1 was *n* = 1222 students and *n* = 419 staff/managers; at T2 the study population was *n* = 1452 students and *n* = 452 staff/managers. A sub-sample of students who smoked daily or occasionally (*n* = 373 at T1 and *n* = 397 at T2) was used to estimate the associations between implementation fidelity and “support to cope with not smoking during school hours” and “smoking cessation assistance”. Likewise, at staff/manager level, the association between implementation fidelity and the “joint workshop before policy implementation” was analyzed with a sub-sample of staff/managers who had attended the workshop and answered the questionnaire (*n* = 184 at T1 and *n* = 134 at T2). The included individuals (N) in each adjusted analysis are shown in Appendix A.

### 2.6. Measures

#### 2.6.1. Implementation Fidelity Measures—Dependent Variables

The implementation fidelity outcome measures were built upon other studies. Inspired by Bast et al. (2016) [24], we defined four STP fidelity domains. The specific items were based on previous studies [19,21,24,25] and normalization process theory [44]. We defined the domains as follows: (1) adherence: familiarity with policy entailments, (2) dose: exposure to smoking during school hours, (3) quality of delivery: enforcement of policy, and (4) participant responsiveness: sense of policy implementation. Unlike constructs that seek to evaluate the comprehensiveness of STPs in general [45], this construct is based on an already defined STP. In other words, the comprehensiveness of the policy was known in advance and measured by adherence.

The domains were assessed using questions tailored to either students or staff/managers and coded as binary variables (implemented = 1 or not = 0). In addition, a total implementation fidelity score was calculated by the sum across fidelity domains (range: 0–4). All variables, including response categories and satisfactory levels of implementation for the binary coding, are shown in Table 2. The satisfactory levels of implementation were decided by the authors based on theoretical discussions and practical experience.

#### 2.6.2. Intervention Activities—Independent Variables

All six intervention activities were assessed using a 5-point Likert scale.

At student level, the assessed intervention activities were smoke-free signage (one item), new school-break facilities (one item), support to cope with not smoking during school hours and smoking cessation assistance (two items). The latter was included as means across the two items (Cronbach’s alpha: 0.881).

At staff/manager level, the assessed intervention activities comprised the joint workshop before policy implementation (five items to measure whether the workshop contributed to a shared understanding, a shared language, as well as new competences and ideas), internalization of fixed procedures for enforcement (one item), and experienced support from NGOs and local municipality (one item). The operationalization of the “joint workshop before policy implementation” was inspired by the Communities of Practice theory [46] and included as means across the five items (Cronbach’s alpha: 0.841).

All intervention activity variables are shown in Table 3.

#### 2.6.3. Context—Confounding Variables

As confounders, we included age, sex, and smoking status at both student and staff/manager levels. At student level, we also included enrolment in the normal or higher vocational education program and main subject area. At staff/manager level, we included special functions in relation to health promotion (e.g., mentor, occupational health and safety representative, part of the school’s health team). Smoking status was also explored as a possible effect-modifier at student level. All confounder variables are defined in detail in Appendix A.

### 2.7. Statistical Analysis

A positive intra-class correlation (ICC) is expected within schools [47], as implementation and experiences with intervention activities are expected to vary less within schools than between schools. To calculate the ICC for binary outcomes, we followed the method proposed by Twisk (2006) [48]. To account for the clustered data structure, we applied generalized linear mixed models with random intercepts at the school level.

We analyzed associations between implementation fidelity (dependent variables) and intervention activities (independent variables) separately for students and staff/managers, using the same procedures. We examined associations between all intervention activity variables and the four dichotomous fidelity outcome variables in univariate and multivariate models using logistic regression, adjusted for all described confounders. Furthermore, we estimated univariate and multivariate linear regression models with the same independent variables and the total implementation fidelity score variable as the outcome variable.

As a sensitivity analysis to detect potential effect-modification, we included “smoking status” as an interaction term with the intervention activities (e.g., smoke-free signage × smoking status) in all models at student level.

Additionally, we conducted all regression analyses with data that excluded the school with (*n* = 102) missing responses on one fidelity variable at T1. As no notable differences were found between the full vs. the reduced data set, the results from the reduced data set are not reported.

Regression analyses were conducted in R Version 4.1.2 using the Lme4 package.

## 3. Results

Table 4 presents the characteristics of the total study population at T1 and T2. The average age of students was 23.5 at T1 and 22.5 at T2, while the smoking prevalence–defined as both daily and occasional smoking–was 30.5% at T1 and 27.3% at T2. The characteristics of the total study population stratified by the seven intervention schools are available in Appendix A, which also contains the student study population stratified by smoking status.

Descriptive results on intervention activities (mean values) at both time points are presented in Table 5. The lowest intervention activity scores were found for “help to cope with not smoking during school hours and smoking cessation assistance” (2.01 at T1; and 1.73 at T2) at student level. While the highest values were found at staff/manager level regarding the joint workshop before policy implementation (3.35) and experienced support from NGOs and local municipality (3.46 at T1; and 3.36 at T2). The means stratified by intervention schools are available in Appendix A.

Table 6 shows the descriptive implementation fidelity results at T1 and T2. At student level, the total implementation fidelity mean was 2.24 at T1 and 2.36 at T2; at staff/manager level, the total implementation fidelity average was 2.80 at T1 and 2.92 at T2. As such, the total implementation fidelity scores increased from T1 to T2 among both students and staff/managers. The results stratified by the intervention schools are presented in Appendix A, which shows that implementation fidelity results varied between schools, with the lowest implementation degrees found at school 1 and school 5.

Table 7 presents the associations at student level between intervention activities and implementation fidelity at T1 and T2, adjusted for all confounders. New school-break facilities were consistently–over time–associated with dose (OR_T1_ = 1.16, 95% CI = 1.03–1.31; OR_T2_ = 1.18, 95% CI = 1.06–1.31), quality of delivery (OR_T1_ = 1.26, 95% CI = 1.02–1.56; OR_T2_ = 1.25, 95% CI = 1.04–1.50), and the total implementation fidelity score (B_T1_ = 0.08, 95% CI = 0.03–0.12; B_T2_ = 0.07, 95% CI = 0.03–0.11). New school-break facilities were only associated with adherence (OR_T1_ = 1.24, 95% CI = 1.05–1.45) at T1, and only with participant responsiveness (OR_T2_ = 1.27, 95% CI = 1.14–1.43) at T2. Smoke-free signage was associated with adherence (OR_T1_ = 1.43, 95% CI = 1.24–1.65; OR_T2_ = 1.39, 95% CI = 1.22–1.58) at both time points. Help to cope with not smoking during school hours and smoking cessation assistance were associated with quality of delivery only at T1 (OR_T1_ = 1.70, 95% CI = 1.02–2.83), which was the only association found for these two items.

Table 8 presents the adjusted associations at staff/manager level between intervention activities and implementation fidelity at T1 and T2. The joint workshop before policy implementation was consistently associated with participant responsiveness (OR_T1_ = 1.43, 95% CI = 1.24–1.65; OR_T2_ = 1.39, 95% CI = 1.22–1.58) and the total implementation fidelity score (B_T1_ = 0.13, 95% CI = 0.02–0.25; B_T2_ = 0.13, 95% CI = 0.004–0.24). Likewise, internalization of fixed enforcement procedures was consistently associated with adherence (OR_T1_ = 1.92, 95% CI = 1.28–2.88; OR_T2_ = 1.57, 95% CI = 1.05–2.34), participant responsiveness (OR_T1_ = 1.59, 95% CI = 1.29–1.97; OR_T2_ = 1.66, 95% CI = 1.33–2.06), and the total implementation fidelity score (B_T1_ = 0.19, 95% CI = 0.13;0.26; B_T2_ = 0.16, 95% CI = 0.13;0.26). No other associations between enforcement procedures or the joint workshop and implementation fidelity were found. Experienced support from NGOs and local municipality was associated only with participant responsiveness (OR_T1_ = 1.51, 95% CI = 1.18–1.95) and with the total implementation fidelity score (B_T1_ = 0.14, 95% CI = 0.05–0.22) at T1.

The ICCs for the models presented in Table 7 and Table 8 were generally low (median = 0.018), which indicates that the results are explained more by individual differences than by school clustering [49].

Appendix A shows the crude analyses, which yield similar results as the adjusted analyses. In addition, the sensitivity analysis available in Appendix A shows that the student-level associations between smoke-free signage and adherence at T1 was positive for smokers (OR_T1_ = 2.12, 95% CI = 1.34–3.35) and negative for non-smokers (OR_T1_ = 0.47, 95% CI = 0.29–0.74, *p* for interaction = 0.001). All other associations between intervention activities and implementation fidelity were similar regardless of smoking status.

## 4. Discussion

### 4.1. Key Findings

The total implementation score increased from the first (T1) to the second time point (T2) at both student and staff/manager level. Three intervention activities of the Smoke-Free Vocational Schools intervention (new school-break facilities, the joint workshop before policy implementation, and internalization of fixed procedures for enforcement) were consistently—i.e., over time—associated with the total implementation fidelity score. The remaining three intervention activities (smoke-free signage, help to cope with not smoking during school hours and smoking cessation assistance, and experienced support from NGOs and local municipality) were only associated with specific implementation fidelity indicators or association was only found at the first time point (T1).

### 4.2. Interpretation of Results

We found that the total implementation fidelity score increased over time at both student and staff/manager levels, which suggests that implementation of the smoke-free school hours policy is gradually becoming part of routine practice. Implementing new practices in organizations takes time. The implementation process ends when the innovation is either abandoned or becomes part of the organizational routine [11,50]. In addition, we found that implementation fidelity was generally higher at the staff/manager level compared with the student level, which suggests that routinization of the policy occurs first at the organizational level. This is plausible as students may only experience the policy as a normal part of everyday life at school once it is implemented by staff/managers [41].

Out of the three intervention activities at student level, new school-break facilities were most strongly—and consistently—associated with implementation fidelity indicators. This suggests that implementing structural (social) alternatives to smoking communities might improve the implementation of the smoke-free school hours policy [40], especially considering that some students smoke during school hours because they are bored [40,51] and some smoke as a social bonding activity [52]. Similar strategies have previously been applied successfully to reduce smoking continuation among occasional smokers at Danish vocational schools [53]. Moreover, the successful Icelandic Prevention Model [54] is an example of how implementing structural alternatives, for example, free access to leisure time activities, can be used as a strategy to reduce substance use among adolescents.

The association between smoke-free signage and adherence (familiarity with the policy entailments) is intuitively understood: the communication works as intended. Furthermore, we found that at T1 the association was stronger for smokers compared with nonsmokers. This is to be expected, as the signs most directly targeted smokers and therefore may be more salient to smokers. The association becomes less strong and non-significant at T2, potentially because the target group pays less attention to the signs over time. Other studies have operationalized visual communication to measure the implementation of STPs [19,32,35]. Yet, this study shows that communication might improve policy implementation, which has also been suggested in previous research [55,56].

At staff/manager level, we found that the joint workshop prior to implementation was associated with both participant responsiveness (sense of policy implementation) and the total implementation score. Other studies have theorized that staff/managers must be adequately prepared for their professional roles and responsibilities in relation to STPs [15,37,39,40,57]. To our knowledge, this is the first study to test—and verify—that a specific workshop activity prepared staff and managers for policy implementation.

Internalization of fixed enforcement procedures was consistently associated with three implementation fidelity indicators. The importance of enforcement has been highlighted many times in literature on STPs [6,7]. Recent research suggests that staff must feel equipped to enforce a policy [15,40] (e.g., know what to do or who to refer to when students violate the rules). This might require a written enforcement strategy in the STP with well-proportioned progressive disciplinary measures [25,39], which was applied in the Smoke-Free Vocational Schools intervention. In this study, we measured internalization, which was associated with both a communicative aspect of implementation (adherence) as well as more general implementation aspects (participant responsiveness and the total implementation fidelity score). Surprisingly, we did not find an association with the actual enforcement (quality of delivery), nor with exposure to smoking during school hours (dose). A possible explanation is that we did not assess the actual enforcement procedures or strategies, which varied between schools.

### 4.3. Operationalization of Policy Implementation Outcomes

In this study, we operationalized the implementation of the smoke-free school hours policy as fidelity domains. As recommended by Martinez et al. (2014) [58] and Glasgow et al. (2013) [59], our aim was to construct a pragmatic/simple outcomes measure. Our measures focused only on cigarette smoking among students, as reducing cigarette smoking was the main concern in the Smoke-Free Vocational Schools intervention [41]. However, a more comprehensive measure could include similar items about, for example, electronic cigarettes or exposure to staff smoking.

While we are confident that we have included domains relevant to capturing implementation fidelity, the operationalization of specific items may be discussed. Below we discuss the operationalization of “participant responsiveness” and “quality of delivery” in detail.

Participant responsiveness was designed to capture the sense of policy implementation. As such, this domain reflects the overall implementation of the policy (as does the total implementation score). At staff/manager level, we applied a validated item from normalization process theory that assesses the general normalization (“do you feel that [the intervention] is currently a normal part of your work?”) [44]. At student level, we simplified the item (“do you sense that people smoke [at this setting]?”). The latter item has been piloted in a similar intervention project at Danish vocational schools to ensure face validity [60] and was considered appropriate by practice stakeholders; however, it has not been thoroughly validated, which is a limitation.

With regard to quality of delivery, we aimed to assess the degree to which enforcement was strict and consistent. In the literature, the operationalization of enforcement varies substantially [6]. For example, enforcement has been measured as smoking visibility [26], the frequency of enforcement [24], or the specific systems/sanctions to monitor smoking behavior [17]. At staff/manager level, we measured enforcement as a crosstab of “exposure to students smoking” and ”frequency of enforcement” to measure whether staff’s enforcement or lack thereof matches the extent to which they see students smoking. In addition, because staff/managers might over-report their consistency in enforcement if asked more directly about their response to students smoking (e.g., how often do you see students smoking and turn a blind eye?), this measure might reduce conformity bias. At student level, similar to other studies [26,31], students were asked about what normally happens when students violate the policy. Although soft sanctions (e.g., help to cope with not smoking during school hours) are more likely to produce desired behavior outcomes compared with hard sanctions (e.g., written warning) [17,18,61,62], we considered either as satisfactory implementation. In this study, students were almost completely unanimous (>90% reported that the rules were enforced), which does not correspond the results from the staff/manager surveys and thus implies questionable validity.

In relation to both participant responsiveness and quality of delivery at student level, we included the response “I don’t know” as satisfactory implementation. This was decided based on the method of exclusion: “I don’t know” does not mean that the student has the impression that smoking was going on during school hours nor that the sanctions were not being enforced. However, we have not validated this decision, which is a shortcoming.

Similar to other studies [19,25], we used different wording for items targeting students or staff/managers, which raises the question of whether we always measured the same phenomena at student and staff/manager levels. With respect to the above discussions, we highlight that our operationalization of the domains “participant responsiveness” and “quality of delivery” at student level might need further scrutiny.

Structured observations were not part of the fidelity measure. As described elsewhere [41], we originally planned to apply this method, as structured observations are often considered more objective and thus appropriate for capturing implementation fidelity [63]. In addition, the method has previously been used to study the implementation of STPs [21,29,32,33,64] by, for example, counting the number of individuals who smoke or the smoke-free signage. Yet, we found the method unreliable to assess exposure to smoking. First, our presence at the schools was only sparse, and second, we were not familiar with the social (smoking) practices or ‘hang-out-spots’ at the schools. As such, a reliable measure would require “frequent onsite assessment” [17] and anthropological insights, which was not feasible. Furthermore, we argue that the way in which smoking is perceived by a specific group constitutes the social norms and is thus more relevant to assess in relation to STP normalization (compared to an objective measure).

### 4.4. Strengths and Limitations

The results must be interpreted in relation to below overall strengths and limitations.

An advantage of this study is the repeated measurements design, which reduces the risk of significant associations occurring at random (due to many regression analyses). An additional study strength is that we included both students and staff/managers to study the implementation of STPs. To our knowledge, this is the first study to apply a representative sample of staff/managers, and we did not find indications of selection bias. Moreover, it is one of the first studies to treat implementation as a dependent variable with a view to understanding how this variable may be improved [14].

As the study is based on Danish vocational schools representing all four main tracks and a large geographical distribution, it is likely that the results can be generalized to Danish vocational schools and similar settings nationally and internationally. As the intervention activities are evidence-based, we expect that the results are also relevant to states/organizations that are in the process of implementing comprehensive STPs in other settings.

Due to the cross-sectional study design, it is not possible to differentiate between cause and effect. As such, inferences about causality in the associations between intervention activities and implementation cannot be made. We included the most important/obvious confounders; however, we did not include all variables that could influence intervention activities and policy implementation. For example, due to the small number of schools, we were not able to include school-level variables.

We were not able to statistically validate the implementation outcomes measure due to the binary structure, which is a shortcoming. In addition, as discussed above, there were limitations in relation to the operationalization of “participant responsiveness” and “quality of delivery” at student level. Although we did not assess the reliability of the implementation fidelity measure, we found indications of reliability, as the instrument ‘behaved’ reasonably across contexts and across time.

From a theoretical point of view, it is relevant to add the intervention activities together in one model as the intervention as a whole is expected to impact implementation fidelity (not only the separate activities). However, this was not feasible due to statistical power. In addition, even though the intervention included several activities [41], we included only the intervention activities that could be measured based on surveys given to all students and staff/managers.

### 4.5. Implications for Research and Practice

To address the need for a construct that specifically reflects STP implementation success, we developed an implementation fidelity measure. Based on the above discussion, we emphasize that (1) surveys (not observations) are appropriate to capture STP implementation fidelity. (2) Applying representative samples of both students and staff/managers is likely to increase the validity of STP implementation measurements. (3) Analyzing the STP implementation over time provides indications as to whether the STP is becoming part of routine practice. A more comprehensive study design could include a longer time frame with more time points and/or follow the same individuals over time (if possible). (4) In this study, we operationalized fidelity measures as binary domains—as well as a total score across domains – to capture implementation outcomes of a specific comprehensive STP. Even though fidelity measures are often intervention-specific [65], we believe our construct can guide future studies on STP implementation given the theoretical underpinnings and pragmatic form. Further research should scrutinize content validity and psychometric properties.

In this study, we quantified associations between intervention activities and policy implementation. However, qualitative research may be applied to explore in depth the processes/mechanisms that drive and shape implementation, which could deepen our understanding of the results. Finally, further research is needed to examine how the implementation of the smoke-free school hours policy impacts students’ smoking behavior and attitudes towards smoking.

This study has direct implications for practice as the results highlight specific actions that are associated with the implementation of the smoke-free school hours policy. For example, in Denmark, the policy has been mandated by law and this study provides knowledge that can aid the schools in their implementation process. Especially establishing new school-break facilities as alternative activities to social smoking might ease implementation. Moreover, preparing staff and managers through a joint workshop as well as equipping them for enforcement can improve implementation.

## 5. Conclusions

We developed implementation fidelity outcomes measures for a comprehensive smoke-free school hours policy. The intervention activities of the Smoke-Free Vocational Schools intervention were associated with implementation outcomes of the policy. At student level, introducing structural alternatives to reduce social smoking is likely to improve the implementation outcomes. At staff/manager level, providing a joint workshop before policy implementation, which clearly describes the purpose and legitimacy of the policy, can increase the implementation. Furthermore, ensuring that staff and managers feel equipped for enforcement is likely to increase the implementation outcomes.

## Figures and Tables

**Figure 1 ijerph-19-12489-f001:**
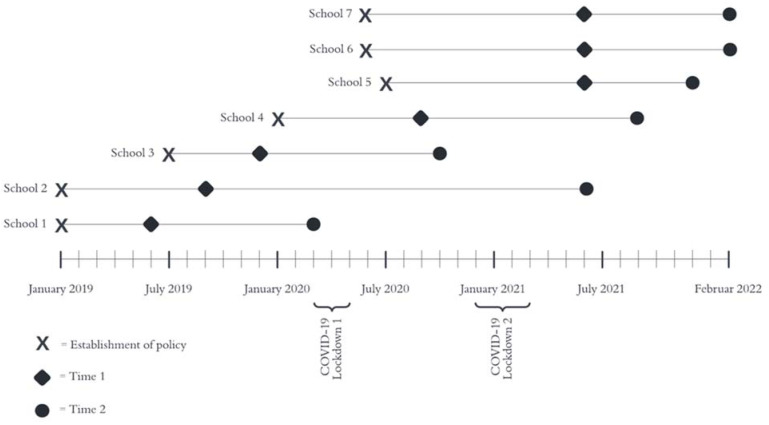
Data collection at Time 1 and Time 2 after the establishment of the smoke-free school hours policy.

**Table 1 ijerph-19-12489-t001:** Intervention activities at each school in the Smoke-Free Vocational Schools intervention, including purpose of activity and target population group.

Intervention Activity	Description	Purpose	Target Population
New school-break facilities	During a participatory workshop, students co-created ideas on how to improve the social environment, and the school received funding to implement ideas (€15,000 per school).	To replace social smoking with locally acceptable activities.	Students
Smoke-free signage	Smoke-free signage (e.g., posters, go-cards) was provided for the schools, and the schools had the option to create their own smoke-free signage (e.g., a smoke-free logo).	To make the policy visible to students (and others).	Students
Help for students to cope with not smoking during school hours	Selected staff and managers participated in a two-day motivational interviewing course, which was tailored to target students enrolled in upper-secondary education. During the course, the participants learned about the social, psychological, and physical aspects of nicotine dependence and how to address the students who were struggling with not smoking during school hours in a positive, communicative way. (Average of *n* = 10 educated staff at each intervention school.)	To provide support to students who find it difficult not to smoke during school hours	Students
Smoking cessation assistance	Smoking cessation courses were offered in collaboration with the local municipality. The courses were offered to both students and staff/managers (separate courses).	To facilitate smoking cessation for students who were motivated to quit smoking.	Students
A joint workshop before policy implementation	A joint workshop for all staff and managers was held to discuss the purpose and legitimacy of the smoke-free school hours policy. At the workshop, the principal presented the school’s motives for adopting the policy as well as the rules for sanctioning and enforcement. Knowledge about organizational change processes and the complexity of nicotine dependence was also presented by a psychologist from a public health NGO. In addition, facilitated group discussions and exercises took place. (*n* = 1 joint workshop at each school.)	To stimulate a shared understanding of why the school is implementing the policy. Additional goals were to develop a shared language and tools that can be used in the implementation process.	Staff/managers
Internalization of fixed enforcement procedures	The schools were obliged to develop school-specific rules for sanctioning and enforcement. The schools were advised to establish rules with a progressive application of disciplinary measures. The rules were then integrated into the schools’ rules of conduct and communicated to all staff and students.	To clearly communicate the rules, so staff and managers know what to do if students violate the policy.	Staff/managers
Support from public health NGOs and local municipality	Throughout the intervention, the schools were supported by both the local municipality and two Danish public health NGOs (the Danish Heart Foundation and the Danish Cancer Society). The NGOs delivered the intervention activities at the schools. Approximately *n* = 5–6 encounters per school in addition to the intervention activities.	To provide support for the schools in the implementation process, when needed.	Staff/managers

**Table 2 ijerph-19-12489-t002:** Implementation fidelity outcome measures—student level and staff/manager level.

Implementation Fidelity Concepts	Definition	Items	Response Categories. Satisfactory Levels of Implementation for the Binary Transformation in Bold
*Student level*			
Adherence	Familiarity with policy entailments	What are the school’s rules on smoking?	We are allowed to smoke everywhere on school property/We are allowed to smoke in designated smoking areas/We are only allowed to smoke outside school premises/**We are not allowed to smoke during the entire school day-neither on school grounds nor outside school grounds**/I don’t know the school’s rules on smoking/The school doesn’t have rules on smoking
Dose	Exposure to smoking during school hours	How often do you see school students smoking during school hours?	Every day or several times a day/Almost daily or a couple of times per week/Circa once a week/**Less than once a week/A few times per month/Rarer/Never**
Quality of delivery	Enforcement of policy	What normally happens when students break the school’s rules on smoking?	**Reprimand or negative sanctioning/Help to cope with not smoking during school hours (e.g., referral to school or municipal smoking cessation counselors)/Other**/Nothing happens/**I don’t know**
Participant responsiveness	Sense of policy implementation	At your school, do you sense that students smoke during school hours?	Yes, on the school premises/Yes, outside the school premises/**No/I don’t know**
*Staff/manager level*			
Adherence	Familiarity with policy entailments	Is it currently permitted for school students to smoke cigarettes during the school day?	Yes/Yes, outside school premises/**No**/I don’t know
Dose	Exposure to smoking during school hours	How often do you see school students smoking during school hours?	Every day or several times a day/Almost daily or a couple of times per week/Circa once a week/**Less than once a week/A few times per month/Rarer/Never**
Quality of delivery	Enforcement of policy	New variable constructed by a crosstab of: How often do you articulate/enforce the smoke-free school hours policy? AND ‘Exposure to smoking during school hours’ construct (Dose)	**Policy is enforced every day/Almost daily/Circa once a week/Less than once a week/A few times per month** AND **Students are seen smoking every day/Almost daily/Circa once a week** **Policy is enforced every day/Almost daily/Circa once a week/Less than once a week/A few times per month** AND **Students are seen smoking less than once a week/A few times per month/Rarer/Never****Policy is never/rarely enforced** AND **Students are seen smoking less than once a week/A few times per month/Rarer/Never** Policy is never/rarely enforced AND Students are seen smoking every day/Almost daily/Circa once a week
Participant responsiveness	Sense of policy implementation	To what extent do you feel that smoke-free school hours are currently a normal part of everyday life at school?	1 = Very little extent, 2 = Little extent, 3 = Neutral, **4 = Large extent, 5 = Very large extent**

**Table 3 ijerph-19-12489-t003:** Intervention activity variables—student level and staff/manager level.

Intervention Activity	Definition	Items	Responses Categories	Timing
*Student level*				
New school-break facilities	Appraisal	Do you agree or disagree with the following statement: After the school established smoke-free school hours, there are a lot of things to do during school breaks	1 = Strongly disagree, 2 = Disagree, 3 = Neutral, 4 = Agree, 5 = Strongly agree	At T1 and T2
Smoke-free signage	Prominence of smoke-free signage	To what extent do you think the smoke-free school hours signage is visible at your school?	1 = Very little extent, 2 = Little extent, 3 = Neutral, 4 = Large extent, 5 = Very large extent	At T1 and T2
Motivational interviewing	Support to cope with not smoking during school hours and smoking cessation assistance	To what extent do you experience that there are staff at this school who can… (1) help you deal with smoking urges during school hours, (2) help you with smoking cessation	1 = Very little extent, 2 = Little extent, 3 = Neutral, 4 = Large extent, 5 = Very large extent	At T1 and T2
Smoking cessation assistance				
*Staff/manager level*				
Joint workshop for all organizational members	A shared understanding, new competences and ideas, and a shared language (mutual engagement, joint enterprise, and shared repertoire)	Do you agree or disagree with the following statements: The joint meeting contributed to… (1) give us a shared understanding of why we have/are going to have smoke-free school hours, (2) give us concrete ideas about how to support students to cope with smoke-free school hours, (3) give me new knowledge about the complexity of nicotine dependence, (4) set in motion a dialogue about smoke-free school hours at the school, (5) give us a shared language, which we use when we talk about smoke-free school hours among employees	1 = Strongly disagree, 2 = Disagree, 3 = Neutral, 4 = Agree, 5 = Strongly agree	Before policy
Internalization of fixed enforcement procedures	Internalization of fixed enforcement procedures	To what extent do you feel equipped to enforce smoke-free school hours? (i.e., know what to do or who to refer to do)	1 = Very little extent, 2 = Little extent, 3 = Neutral, 4 = Large extent, 5 = Very large extent	At T1 and T2
Experienced support from NGOs and local municipality	Appraisal	Do you agree or disagree with the following statement: External help from the municipality or The Danish Cancer Society or The Danish Heart Foundation is supportive in relation to our work with smoke-free school hours	1 = Strongly disagree, 2 = Disagree, 3 = Neutral, 4 = Agree, 5 = Strongly agree	At T1 and T2

**Table 4 ijerph-19-12489-t004:** Study population characteristics at T1 and T2.

	Students T1	Students T2	Staff/Managers T1	Staff/Managers T2
Individuals, N (%)	1222 (100)	1452 (100)	419 (100)	452 (100)
Age, range	15–63	15–66	20–67	17–76
Age, mean (SD)	23.5 (9.3)	22.5 (8.5)	48.6 (9.8)	46.9 (10.2)
Male gender, %	59.0	58.5	42.7	42.7
Smoking prevalence *, %	30.5	27.3	12.9	12.4
Educational track, %				
Care, health, and pedagogy	29.2	28.2	NA	NA
Administration, commerce, and business service	17.9	27.7	NA	NA
Food, agriculture, and hospitality	5.8	7.0	NA	NA
Technology, construction, and transportation	47.1	37.0	NA	NA
Educational level, %				
Vocational school-normal	74.0	66.0	NA	NA
Vocational school-higher	26.0	34.0	NA	NA
School position, %				
Manager	NA	NA	11.0	5.7
Teacher	NA	NA	62.8	65.9
Counsellor	NA	NA	6.7	6.9
Administrative	NA	NA	11.7	11.0
Other positions	NA	NA	7.9	10.6
Special function in relation to health promotion **, %	NA	NA	47.5	47.0

*** Smoking prevalence is defined as both daily and occasional smoking. ** At both time points, the special functions mostly included ‘contact teachers’ (approx. 30%) i.e., a person who the students can contact in relation to both educational goals and personal issues.

**Table 5 ijerph-19-12489-t005:** Descriptive results regarding intervention activities at T1 and T2—mean values.

Intervention Activities *	Students T1 Mean (SD)	Students T2 Mean (SD)	Staff/Managers T1 Mean (SD)	Staff/Managers T2 Mean (SD)
New school-break facilities	2.82 (±1.1)	2.91 (±1.2)	NA	NA
Smoke-free signage	2.58 (±1.3)	2.52 (±1.3)	NA	NA
Help to deal with not smoking during school hours and smoking cessation assistance	2.01 (±1.2)	1.73 (±1.0)	NA	NA
Joint workshop before policy implementation	NA	NA	3.35 (±0.9)	3.35 (±0.9)
Internalization of fixed enforcement procedures	NA	NA	3.07 (±1.1)	2.98 (±1.0)
Experienced support from NGOs and local municipality	NA	NA	3.46 (±0.9)	3.36 (±0.8)

* All intervention activities were assessed on Likert scales from 1–5.

**Table 6 ijerph-19-12489-t006:** Descriptive results regarding implementation fidelity at T1 and T2—proportion of ‘implemented’.

	Students T1 %	Students T2 %	Staff/Managers T1 %	Staff/Managers T2 %
Adherence	87.3	86.8	93.6	93.8
Dose	28.2	32.7	65.4	68.6
Quality of delivery	91.1	92.6	70.6	62.8
Participant responsiveness	25.3	28.4	51.1	67.7
Total implementation fidelity, mean (SD) *	2.24 (0.8)	2.36 (0.8)	2.80 (0.8)	2.92 (0.7)

* Total implementation fidelity is the sum across fidelity measures (range: 0–4).

**Table 7 ijerph-19-12489-t007:** **Student level** associations between intervention activities and implementation fidelity of the smoke-free school hours policy, across T1 and T2, adjusted for age, sex, smoking status, educational level enrolled and main subject area.

	Odds Ratio (OR) with 95% Confidence Interval (95% CI) *p*-Value				Linear Effect with 95% CI and *p*-Value
Intervention Activities	Adherence	Dose	Quality of Delivery	Participant Responsiveness	Total Implementation Fidelity
*Student level-time 1 (T1)*					
New school-break facilities	**1.24 [1.05–1.45] 0.008**	**1.16 [1.03–1.31] 0.009**	**1.26 [1.02–1.56] 0.002**	1.06 [0.94–1.20] 0.290	**0.08 [0.03–0.12] 0.000**
Smoke-free signage	**1.43 [1.24–1.65] 0.000**	0.94 [0.86–1.04] 0.291	1.15 [0.96–1.37] 0.119	0.93 [0.87–1.04] 0.223	0.02 [−0.008–0.06] 0.145
Help to cope with not smoking during school hours and smoking cessation assistance	1.06 [0.77–1.46] 0.687	1.04 [0.86–1.26] 0.665	**1.70 [1.02–2.83] 0.037**	1.05 [0.84–1.30] 0.654	0.06 [−0.008–0.12] 0.082
*Student level-time 2 (T2)*					
New school-break facilities	0.89 [0.77–1.03] 0.139	**1.18 [1.06–1.31] 0.001**	**1.25 [1.04–1.50] 0.017**	**1.27 [1.14–1.43] 0.000**	**0.07 [0.03–0.11] 0.000**
Smoke-free signage	**1.39 [1.22–1.58] 0.000**	0.92 [0.84–1.00] 0.06	1.14 [0.97–1.35] 0.103	0.94 [0.86–1.03] 0.175	0.01 [−0.01–0.05] 0.255
Help to cope with not smoking during school hours and smoking cessation assistance *	1.02 [0.72–1.46] 0.881	0.89 [0.71–1.13] 0.364	0.99 [0.59–1.65] 0.978	1.05 [0.84–1.32] 0.631	−0.003 [−0.08–0.07] 0.934

* The analyses for this intervention activity were not adjusted for smoking status (sub-group of smokers only).

**Table 8 ijerph-19-12489-t008:** **Staff/manager level** associations between intervention activities and implementation fidelity of the smoke-free school hours policy, across T1 and T2. The models are adjusted for age, sex, smoking status, and if they had a special function in relation to health promotion.

	Odds Ratio (OR) with 95% Confidence Interval (95% CI) *p*-Value				Linear Effect with 95% CI and *p*-Value
Intervention Activities	Adherence	Dose	Quality of Delivery	Participant Responsiveness	Total Implementation Fidelity
*Staff/manager level-Time 1 (T1)*					
Joint workshop before policy implementation	1.45 [0.50–4.21] 0.489	0.95 [0.65–1.40] 0.821	1.17 [0.81–1.68] 0.394	**1.66 [1.13–2.46] 0.010**	**0.13 [0.02–0.25] 0.022**
Internalization of fixed enforcement procedures	**1.92 [1.28–2.88] 0.001**	1.09 [0.89–1.33] 0.402	1.17 [0.95–1.44] 0.131	**1.59 [1.29–1.97] 0.000**	**0.19 [0.13–0.26] 0.000**
Experienced support from NGOs and local municipality	1.08 [0.69–1.68] 0.702	1.09 [0.86–1.39] 0.442	1.19 [0.93–1.51] 0.150	**1.51 [1.18–1.95] 0.001**	**0.14 [0.05–0.22] 0.001**
*Staff/manager level-Time 2 (T2)*					
Joint workshop before policy implementation	1.46 [0.46–4.65] 0.514	1.03 [0.65–1.64] 0.869	1.10 [0.69–1.73] 0.679	**1.73 [1.08–2.76] 0.021**	**0.13 [0.004–0.24] 0.05**
Internalization of fixed enforcement procedures	**1.57 [1.05–2.34] 0.023**	1.17 [0.94–1.45] 0.140	1.05 [0.86–1.28] 0.617	**1.66 [1.33–2.06] 0.000**	**0.16 [0.10–0.23] 0.000**
Experienced support from NGOs and local municipality	0.80 [0.48–1.31] 0.383	1.01 [0.77–1.33] 0.926	1.12 [0.87–1.45] 0.352	1.09 [0.83–1.43] 0.504	0.04 [−0.05–0.12] 0.421

## Data Availability

Anonymized data are available upon reasonable request.

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
