# Peer review of "Intervention Activities Associated with the Implementation of a Comprehensive School Tobacco Policy at Danish Vocational Schools: A Repeated Cross-Sectional Study"

_ijerph, 2022, doi:10.3390/ijerph191912489_

Round 1

Reviewer 1 Report

The manuscript entitled, “Intervention Activities Associated with the Implementation of a Comprehensive School Tobacco Policy at Danish Vocational Schools: A Repeated Cross-Sectional Study” investigates the association between pieces of an intervention to increase fidelity of a school-based smokefree policy and program fidelity. Primary strengths of the paper are the study design, timeliness of the topic, and comprehensive assessment of both students and school staff. My specific comments are listed below. Thank you for the opportunity to review this interesting research.

Specific comments:

From the abstract, it is not clear that a primary aim of the study is to develop implementation outcome measures. Consider adding this to the abstract. This consistency should be increased throughout the paper.

The authors should consider adjusting the introduction to focus more specifically on impacting fidelity. Doing so may better set up the study.

Consider adding a statement in the design section noting the rationale for using a repeated cross-sectional design in this particular instance.

Check methods sections for unnecessary redundancy with the published protocol paper.

Consider a more robust discussion of the implications of developing implementation outcomes measures. Again, since this is a primary aim of the study, it necessitates greater attention throughout the manuscript. This will also help with dissemination and understanding how to best utilize the measures in future research and programs.

Author Response

Author response: We appreciate the time and effort you dedicated to providing feedback on our manuscript.  We incorporated most of your suggestions. Please see below in blue for a point-by-point response to the comments. The line numbers refer to the revised manuscript file with track changes.

The manuscript entitled, “Intervention Activities Associated with the Implementation of a Comprehensive School Tobacco Policy at Danish Vocational Schools: A Repeated Cross-Sectional Study” investigates the association between pieces of an intervention to increase fidelity of a school-based smoke-free policy and program fidelity. Primary strengths of the paper are the study design, timeliness of the topic, and comprehensive assessment of both students and school staff. My specific comments are listed below. Thank you for the opportunity to review this interesting research.

Author response: Thank you for this comment. We are pleased with your overall assessment of our study and your acknowledgment of the study’s strengths.

From the abstract, it is not clear that a primary aim of the study is to develop implementation outcome measures. Consider adding this to the abstract. This consistency should be increased throughout the paper.

Author response: Thank you for the opportunity to improve the manuscript with your valid point. We have included the development of implementation outcomes measures as a subordinate aim and increased the consistency throughout the paper. Specifically:

·         We added sentences in the abstract, which describe the aim and its implication, lines: 21-22 and 23-24, and 39-40.

·         We included the development of implementation outcome measures as an additional aim under Aim, line 127

·         We included a more robust discussion of the implications of the implementation outcomes measure, including how to best utilize the measures in future research. Lines: 543-552

·         Lastly, we have included a sentence in the Conclusion to also address the secondary aim of developing implementation outcome measures. Line: 570-571

The authors should consider adjusting the introduction to focus more specifically on impacting fidelity. Doing so may better set up the study.

Author response: Thank you for this comment. We agree that the concept of implementation fidelity should be described in the Introduction. We added a paragraph that describes the fidelity literature in general and specifically related to school tobacco policies, lines 81-86.

Consider adding a statement in the design section noting the rationale for using a repeated cross-sectional design in this particular instance.

Author response: Thank you for this addition. We have included the rationale for using a repeated cross-sectional design under Study Design, lines: 169-171.

Check methods sections for unnecessary redundancy with the published protocol paper.

Author response: Thank you for this consideration. We acknowledge the need to avoid unnecessary redundancy between papers. However, we would also like to accommodate that readers get enough information, so they do not need to read our study protocol to understand the present study. We have closely re-read the method section and we believe the descriptions are needed in the current study.

Consider a more robust discussion of the implications of developing implementation outcomes measures. Again, since this is a primary aim of the study, it necessitates greater attention throughout the manuscript. This will also help with dissemination and understanding how to best utilize the measures in future research and programs.

Author response: Again, thank you for this sharp observation. We believe we have accommodated this comment by adding the development of an implementation outcomes measure as an aim and ensured the consistency throughout the paper (see above).

Reviewer 2 Report

Title: Intervention Activities Associated with the Implementation of a Comprehensive School Tobacco Policy at Danish Vocational Schools: A Repeated Cross‐Sectional Study

Overall: This is a well-articulated and well-described study with a strong rationale and findings with high public health relevance. Comments below are intended to help improve the quality of the manuscript. Note that, in general, the paper requires careful proofreading and editing.

Abstract: Well-written; however, there is limited space devoted to reporting the results and articulating the implications/conclusions. Suggest revising to address this.

Introduction: Well-written, providing a solid rationale for the study based on the literature and clear gaps the study aims to address.

One suggestion: consider providing a bit more context on the tobacco use and tobacco control policy landscape in Denmark more broadly? As one example, providing the smoking prevalence of vocational college students in itself is not terribly informative – what is the smoking prevalence overall and relative to a similar age group in general?

Methods: Well-described overall.

Timing of assessments: “The first time point was at least five months after the policy had been established, while the second time point was at least 14 months after policy establishment. The exact timing of data collections is shown in Table 2. Vocational schools in Denmark were closed due to COVID‐19 from March 2020 to May 2020 and from end of December 2020 to March 2021. Data collection was not executed during lockdowns.” Could Table 2 be converted to a figure so that it is slightly clearer how COVID disrupted data collection? For example, a timeline that showed assessment at each school by month and year (from June 2019 to February 2022), with the timeline clearly indicating when COVID-related lockdowns occurred (December 2020 to March 2021) and then showing the execution of the first and second assessments by school? I suggest this, as I initially had concern about how much variability there was in timing but then tried to contextualize each school within the context of COVID.

Also: “The joint workshop was instead surveyed shortly after the meeting had taken place, which was before the school had established the policy.” – Provide some parameters for “shortly after” and for “before…policy”?

Section 2.4 seems out of order. It seems like 2.5 and then 2.6 should be reported and then 2.4 could be reported right before or as part of the analysis section. Or could be reported right after 2.5.

The measures in section 2.6.2 are unclear – please be more specific regarding what is assessed with the items? A reader should be able to interpret the study without the appendices.

Results: Well-presented overall.

Related to the comment above (RE 2.6.2 measures), the interpretation of Table 5 is unclear without the appendix.

Discussion: Generally well-written.

First line: “The total implementation score increased over time at both student and staff/manager level.” Unless I missed it, I didn’t see a statistical test of change over time. Where is this data? If it is not included, why not? If it is included, be sure that it is clear. Where this is addressed in the analysis section of the methods is also unclear. Perhaps the use of the phrase “over time” is the problematic point here – there are only 2 timepoints so more precise, clear language might be “at both timepoints”.

I think that it is important to note that the overall fidelity implementation scores for staff/managers was really a reflection of the participant responsiveness component; otherwise, the interpretation of the results is misleading. Similarly, I think it is crucial to note that the activity that carried implementation fidelity among students was the school break facility component.

I also suggest contextualizing this study within the broader Denmark context would help with interpretability.

Author Response

Author response: We appreciate the time and effort you dedicated to providing feedback on our manuscript.  We incorporated your suggestions. Please see below in blue for a point-by-point response to the comments. The line numbers refer to the revised manuscript file with track changes.

Title: Intervention Activities Associated with the Implementation of a Comprehensive School Tobacco Policy at Danish Vocational Schools: A Repeated Cross‐Sectional Study

Overall: This is a well-articulated and well-described study with a strong rationale and findings with high public health relevance. Comments below are intended to help improve the quality of the manuscript. Note that, in general, the paper requires careful proofreading and editing.

Author response: Thank you for this comment. We are pleased with your overall assessment of our study. We have reread the manuscript for further proofreading.

Abstract: Well-written; however, there is limited space devoted to reporting the results and articulating the implications/conclusions. Suggest revising to address this.

Author response: Thank you for this comment. We have included sentences about the development of implementation outcome measures in the abstract as well as its implications for future research. Lines: 21-22, 23-24, and 38-40.

 Introduction: Well-written, providing a solid rationale for the study based on the literature and clear gaps the study aims to address.

One suggestion: consider providing a bit more context on the tobacco use and tobacco control policy landscape in Denmark more broadly? As one example, providing the smoking prevalence of vocational college students in itself is not terribly informative – what is the smoking prevalence overall and relative to a similar age group in general?

Author response: Thank you for the opportunity to improve our paper with this suggestion. We have included context on the tobacco use and tobacco control policy landscape in Denmark, including smoking prevalence for the general population and for similar age groups. Lines: 50-52 and 120-123.

Methods: Well-described overall.

Timing of assessments: “The first time point was at least five months after the policy had been established, while the second time point was at least 14 months after policy establishment. The exact timing of data collections is shown in Table 2. Vocational schools in Denmark were closed due to COVID‐19 from March 2020 to May 2020 and from end of December 2020 to March 2021. Data collection was not executed during lockdowns.” Could Table 2 be converted to a figure so that it is slightly clearer how COVID disrupted data collection? For example, a timeline that showed assessment at each school by month and year (from June 2019 to February 2022), with the timeline clearly indicating when COVID-related lockdowns occurred (December 2020 to March 2021) and then showing the execution of the first and second assessments by school? I suggest this, as I initially had concern about how much variability there was in timing but then tried to contextualize each school within the context of COVID.

Author response: Thank you for this comment – that is a great idea. We converted table 2 to a figure, which gives a nice overview of the data collection timing. Page 7, below from line 187.

Also: “The joint workshop was instead surveyed shortly after the meeting had taken place, which was before the school had established the policy.” – Provide some parameters for “shortly after” and for “before…policy”?

Author response: Thank you for this sharp observation, which we had missed. We have included the parameters for when the joint workshop took place and when it was surveyed. Lines: 184-186.

Section 2.4 seems out of order. It seems like 2.5 and then 2.6 should be reported and then 2.4 could be reported right before or as part of the analysis section. Or could be reported right after 2.5.

Author response: Thank you for this suggestion. We reversed the order of the sections – and moved Section 2.4 after 2.5.

 The measures in section 2.6.2 are unclear – please be more specific regarding what is assessed with the items? A reader should be able to interpret the study without the appendices.

Author response: Thank you for this important point. We included the table in main text (instead of an appendix). Page 10-11, below line 281.

Results: Well-presented overall.

Related to the comment above (RE 2.6.2 measures), the interpretation of Table 5 is unclear without the appendix.

Author response: Thank you for this comment. We believe we have accommodated your suggestion by adding the table, which shows all intervention activity variables, in section 2.6.2.

Discussion: Generally well-written.

First line: “The total implementation score increased over time at both student and staff/manager level.” Unless I missed it, I didn’t see a statistical test of change over time. Where is this data? If it is not included, why not? If it is included, be sure that it is clear. Where this is addressed in the analysis section of the methods is also unclear. Perhaps the use of the phrase “over time” is the problematic point here – there are only 2 timepoints so more precise, clear language might be “at both timepoints”.

Author response: Thank you for this comment. We have rephrased the sentence to make it precise: “The total implementation scores increased from first (T1) to second time point (T2) at both student and staff/manager level”. Line 388.

I think that it is important to note that the overall fidelity implementation scores for staff/managers was really a reflection of the participant responsiveness component; otherwise, the interpretation of the results is misleading.

Author response: Thank you for this good point, which we have incorporated in the Discussion, lines: 459-460.

Similarly, I think it is crucial to note that the activity that carried implementation fidelity among students was the school break facility component.

Author response: Thank you for this comment. We already mention: “Out of the three intervention activities at student level, new school-break facilities were most strongly – and consistently – associated with implementation fidelity indicators” (lines 407-409). Also, for implications for practice, we emphazise that “Especially establishing new school-break facilities as alternative activities to social smoking might ease implementation” (lines: 565-566). Furthermore, in the discussion we have added emphasis on this activity: “At student level, introducing structural alternatives to reduce social smoking is likely to improve the implementation outcomes” (lines: 573-574).   

I also suggest contextualizing this study within the broader Denmark context would help with interpretability.

Author response: Thank you for this comment. As described, we have now added information on the Danish tobacco control landscape in the Introduction. Furthermore, we have added a description of how the results can be used in practice in Denmark under Implications for Research and Practice, lines: 563-565. We hope that these additions will assist with the interpretability to the Danish context.

Reviewer 3 Report

This is a very interesting article, which addresses a very important issue. It makes a good introduction and clearly formulates the objective of the research.

The Results are interesting and well presented. The Discussion is adequate, although more research with which to compare the Results could be sought.

The section on Strengths and limitations of the research is very interesting.

The Conclusions could be improved. Too short. As for the bibliographical references, they are scarce and should be expanded.

1.-  It evaluates the impact of tobacco policies in Denmark in 7 vocational schools.

2.- This is a relevant topic, as it deals with an important issue: health-education-adolescence. It makes a good introduction and clearly formulates the objective of the research. It differs from other research because it evaluates the impact of the policies at two points in time: 5 months and 14 months after their implementation.

3.- The methodology is adequate. The results are interesting and well presented. The Discussion is adequate, although one could look for more research with which to compare the Results. The section on the strengths and limitations of the research is very interesting.

4.- The Conclusions could be improved. They are not consistent and are too short. They do not respond to the objective of the research.

5.- As for the bibliographical references, they are scarce. They should be expanded.

Author Response

Author response: We appreciate the time and effort you dedicated to providing feedback on our manuscript.  We incorporated most of your suggestions. Please see below in blue for a point-by-point response to the comments. The line numbers refer to the revised manuscript file with track changes.

This is a very interesting article, which addresses a very important issue. It makes a good introduction and clearly formulates the objective of the research.

Author response: Thank you for your overall positive assessment of our study.

The Results are interesting and well presented. The Discussion is adequate, although more research with which to compare the Results could be sought.

Author response: Thank you for this comment. We have further developed the Discussion, lines: 459-460, 543-552, and 563-565.

The section on Strengths and limitations of the research is very interesting.

Author response: Thank you. We are pleased that you found this discussion relevant and interesting.

The Conclusions could be improved. Too short.

Author response: Thank you for this comment. We have rephrased the Discussion to include more information, lines: 570-578.

As for the bibliographical references, they are scarce and should be expanded.

Author response: Thank you for this comment. We now have 65 bibliographical references. Thus, we have added three citations.

1.-  It evaluates the impact of tobacco policies in Denmark in 7 vocational schools.

2.- This is a relevant topic, as it deals with an important issue: health-education-adolescence. It makes a good introduction and clearly formulates the objective of the research. It differs from other research because it evaluates the impact of the policies at two points in time: 5 months and 14 months after their implementation.

Author response: Thank you for your consideration. We are pleased that the overall design and introduction are clearly described.

3.- The methodology is adequate. The results are interesting and well presented. The Discussion is adequate, although one could look for more research with which to compare the Results. The section on the strengths and limitations of the research is very interesting.

Author response: Thank you for this comment. We have described (above) the alterations made to the Discussion, which we hope will accommodate this comment.

4.- The Conclusions could be improved. They are not consistent and are too short. They do not respond to the objective of the research.

Author response: Thank you for this comment. Please see above how we accommodated this comment and extended the Conclusion.

5.- As for the bibliographical references, they are scarce. They should be expanded.

Author response: Thank you for this comment. Please see above how we accommodated this comment and added three bibliographical references.